# WORLDMEM: Long-term Consistent World Simulation with Memory

**Zeqi Xiao**[1]   **Yushi Lan**[1]   **Yifan Zhou**[1]   **Wenqi Ouyang**[1]
**Shuai Yang**[2]   **Yanhong Zeng**[3]   **Xingang Pan**[1]

[1]S-Lab, Nanyang Technological University,
[2]Wangxuan Institute of Computer Technology, Peking University
[3]Shanghai AI Laboratory

{zeqi001, yushi001, yifan006, wenqi.ouyang, xingang.pan}@ntu.edu.sg
williamyang@pku.edu.cn, zengyh1900@gmail.com

## Abstract

World simulation has gained increasing popularity due to its ability to model virtual environments and predict the consequences of actions. However, the limited temporal context window often leads to failures in maintaining long-term consistency, particularly in preserving 3D spatial consistency. In this work, we present WORLD-MEM, a framework that enhances scene generation with a memory bank consisting of memory units that store memory frames and states (*e.g.*, poses and timestamps). By employing state-aware memory attention that effectively extracts relevant information from these memory frames based on their states, our method is capable of accurately reconstructing previously observed scenes, even under significant viewpoint or temporal gaps. Furthermore, by incorporating timestamps into the states, our framework not only models a static world but also captures its dynamic evolution over time, enabling both perception and interaction within the simulated world. Extensive experiments in both virtual and real scenarios validate the effectiveness of our approach. Project page at https://xizaoqu.github.io/worldmem.

## 1   Introduction

World simulation has gained significant attention for its ability to model environments and predict the outcomes of actions (Bar et al., 2024; Decart et al., 2024; Alonso et al., 2025; Feng et al., 2024; Parker-Holder et al., 2024; Valevski et al., 2024). Recent advances in video diffusion models have further propelled this field, enabling high-fidelity rollouts of potential future scenarios based on user actions, such as navigating through an environment or interacting with objects. These capabilities make world simulators particularly promising for applications in autonomous navigation (Feng et al., 2024; Bar et al., 2024) and as viable alternatives to traditional game engines (Decart et al., 2024; Parker-Holder et al., 2024).

Despite these advances, a fundamental challenge remains: the limited probing horizon. Due to computational and memory constraints, video generative models operate within a fixed context window and are unable to condition on the full sequence of past generations. Consequently, most existing methods simply discard previously generated content, leading to a critical issue of world inconsistency, which is also revealed in Wang et al. (2025). As illustrated in Figure 1(a), when the camera moves away and returns, the regenerated content diverges from the earlier scene, violating the coherence expected in a consistent world.

A natural solution is to maintain an external memory that stores and retrieves relevant historical information outside the generative loop. While intuitive, formulating such a memory mechanism is

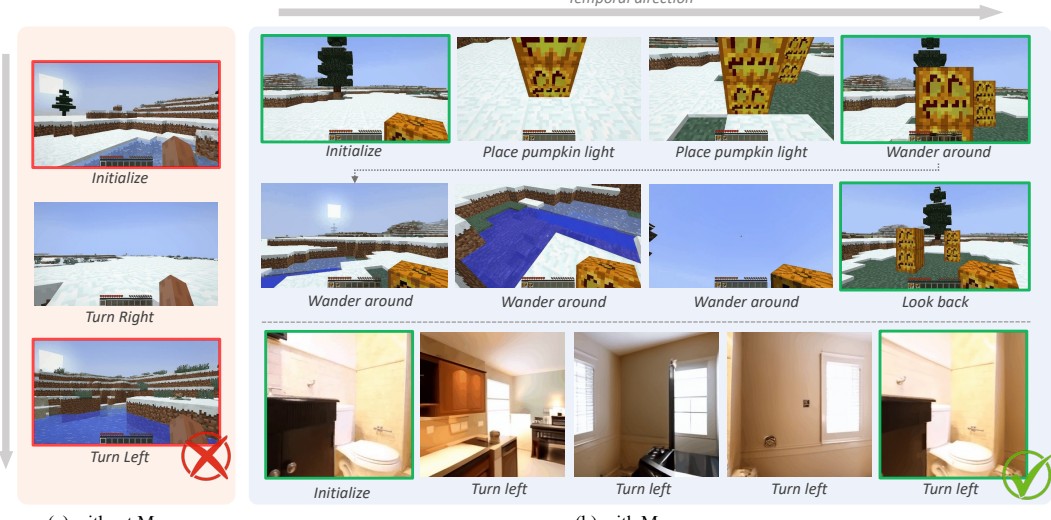

Figure 1: **WORLDMEM enables long-term consistent world generation with an integrated memory mechanism.** (a) Previous world generation methods typically face the problem of inconsistent world due to limited temporal context window size. (b) WORLDMEM empowers the agent to explore diverse and consistent worlds with an expansive action space, *e.g.*, crafting environments by placing objects like pumpkin light or freely roaming around. Most importantly, after exploring for a while and glancing back, we find the objects we placed are still there, with the inspiring sight of the light melting the surrounding snow, testifying to the passage of time. Red and green boxes indicate scenes that should be consistent.

non-trivial. A direct approach might involve explicit 3D scene reconstruction to preserve geometry and detail. However, 3D representations are inflexible in dynamic and evolving environments and are prone to loss of detail, especially for large, unbounded scenes (Wu et al., 2025a).

Instead, we argue that geometry-free representations offer a more flexible solution. These representations, however, pose their own challenges – particularly in balancing detail retention with memory scalability. For example, implicit approaches like storing abstract features via LoRA modules (Hong et al., 2024) offer compactness but lose visual fidelity and spatial specificity. Some recent works represent visual scenes as discrete tokens encoding fine-grained visual information (Sajjadi et al., 2022; Jiang et al., 2025), but they are limited by a fixed token and struggle to capture the complexity of diverse and evolving environments. To address this issue, we observe that for generating the immediate future, only a small subset of historical content is typically relevant. Based on this, we propose a token-level *memory bank* that stores all previously generated latent tokens, and retrieves a targeted subset for each generation step based on relevance.

Conditioning on the retrieved memory requires spatial-temporal reasoning. In contrast to prior work where memory aids local temporal smoothness (Zheng et al., 2024a) or semantic coherence (Wu et al., 2025b; Rahman et al., 2023), long-term world simulation demands reasoning over large spatiotemporal gaps, *e.g.*, memory and query may differ in viewpoint and time, and retain exact scenes with detail. To facilitate this reasoning, we propose augmenting each memory unit with explicit state cues, including spatial location, viewpoint, and timestamp. These cues serve as anchors for reasoning and are embedded as part of the query-key attention mechanism. Through this state-aware attention, our model can effectively reason the current frame with past observations, facilitating accurate and coherent generation. Importantly, such a design leverages standard attention architectures, enabling it to scale naturally with modern hardware and model capacity.

Motivated by this idea, we build our approach, WORLDMEM, on top of the Conditional Diffusion Transformer (CDiT) (Peebles and Xie, 2023) and the Diffusion Forcing (DF) paradigm (Chen et al., 2025), which autoregressively generates first-person viewpoints conditioned on external action signals. As discussed above, at the core of WORLDMEM is a memory mechanism composed of a memory bank and memory attention. To ensure efficient and relevant memory retrieval from the bank, we introduce a confidence-based selection strategy that scores memory units based on field-of-view

(FOV) overlap and temporal proximity. In the memory attention, the latent tokens being generated act as queries, attending to the memory tokens (as keys and values) to incorporate relevant historical context. To ensure robust correspondence across varying viewpoints and time gaps, we enrich both queries and keys with state-aware embeddings. A *relative embedding* design is introduced to ease the learning of spatial and temporal relationships. This pipeline enables precise, scalable reasoning over long-range memory, ensuring consistency in dynamic and evolving world simulations.

We evaluate WORLDMEM on a customized Minecraft benchmark (Fan et al., 2022) and on RealEstate10K (Zhou et al., 2018). The Minecraft benchmark includes diverse terrains (*e.g.*, plains, savannas, and deserts) and various action modalities (movement, viewpoint control, and event triggers), which is a wonderful environment for idea verification. Extensive experiments show that WORLDMEM significantly improves 3D spatial consistency, enabling robust viewpoint reasoning and high-fidelity scene generation, as shown in Figure 1(b). Furthermore, in dynamic environments, WORLDMEM accurately tracks and follows evolving events and environment changes, demonstrating its ability to both perceive and interact with the generated world. We hope our promising results and scalable designs will inspire future research on memory-based world simulation.

## 2   Related Work

**Video diffusion model.** With the rapid advancement of diffusion models (Song et al., 2020; Peebles and Xie, 2023; Chen et al., 2025), video generation has made significant strides (Wang et al., 2023a,b; Chen et al., 2023; Guo et al., 2023; OpenAI, 2024; Jin et al., 2024; Yin et al., 2024). The field has evolved from traditional U-Net-based architectures (Wang et al., 2023a; Chen et al., 2023; Guo et al., 2023) to Transformer-based frameworks (OpenAI, 2024; Ma et al., 2024; Zheng et al., 2024b), enabling video diffusion models to generate highly realistic and temporally coherent videos. Recently, autoregressive video generation (Chen et al., 2025; Kim et al., 2024; Henschel et al., 2024) has emerged as a promising approach to extend video length, theoretically indefinitely. Notably, Diffusion Forcing (Chen et al., 2025) introduces a per-frame noise-level denoising paradigm. Unlike the full-sequence paradigm, which applies a uniform noise level across all frames, per-frame noise-level denoising offers a more flexible approach, enabling autoregressive generation.

**Interactive world simulation.** World simulation aims to model an environment by predicting the next state given the current state and action. This concept has been extensively explored in the construction of world models (Ha and Schmidhuber, 2018b) for agent learning (Ha and Schmidhuber, 2018a; Hafner et al., 2019, 2020; Hu et al., 2023; Beattie et al., 2016; Yang et al., 2023). With advances in video generation, high-quality world simulation with robust control has become feasible, leading to numerous works focusing on interactive world simulation (Bar et al., 2024; Decart et al., 2024; Alonso et al., 2025; Feng et al., 2024; Parker-Holder et al., 2024; Valevski et al., 2024; Yu et al., 2025c,a,b). These approaches enable agents to navigate generated environments and interact with them based on external commands.

However, due to context window limitations, such methods discard previously generated content, leading to inconsistencies in the simulated world, particularly in maintaining 3D spatial coherence.

**Consistent world simulation.** Ensuring the consistency of a generated world is crucial for effective world simulation Wang et al. (2025). Existing approaches can be broadly categorized into two types: geometric-based and geometric-free. The geometric-based methods explicitly reconstruct the generated world into a 3D/4D representation (Liu et al., 2024; Gao et al., 2024; Wang and Agapito, 2024; Ren et al., 2025; Yu et al., 2024b,a; Liang et al., 2024). While this strategy can reliably maintain consistency, it imposes strict constraints on flexibility: Once the world is reconstructed, modifying or interacting with it becomes challenging. Geometric-free methods focus on implicit learning. Methods like Alonso et al. (2025); Valevski et al. (2024) ensure consistency by overfitting to predefined scenarios (*e.g.*, specific CS:GO or DOOM maps), limiting scalability. StreamingT2V (Henschel et al., 2024) maintains long-term consistency by continuing on both global and local visual contexts from previous frames, while SlowFastGen (Hong et al., 2024) progressively trains LoRA (Hu et al., 2022) modules for memory recall. However, these methods rely on abstract representations, making accurate scene reconstruction challenging. In contrast, our approach retrieves information from previously generated frames and their states, ensuring world consistency without overfitting to specific scenarios.

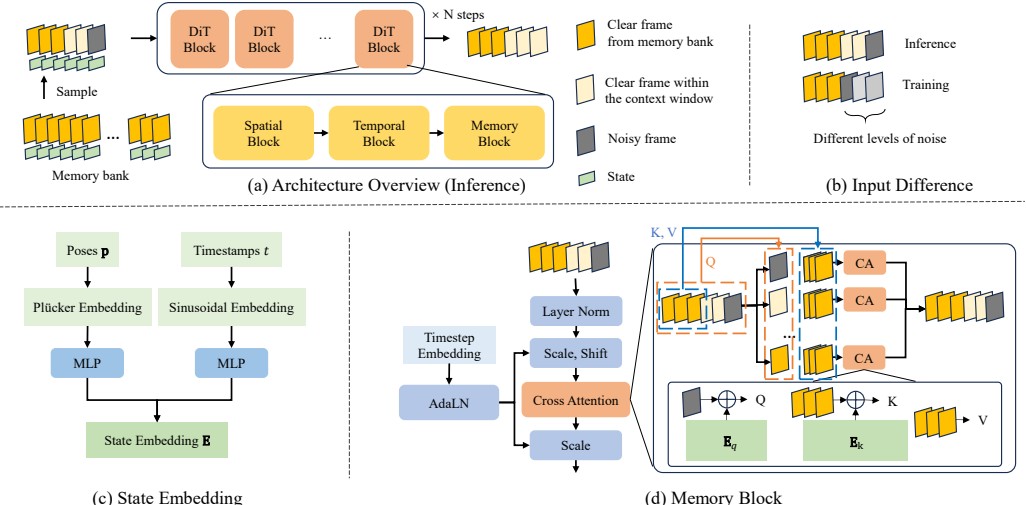

Figure 2: **Comprehensive overview of WORLDMEM.** The framework comprises a conditional diffusion transformer integrated with memory blocks, with a dedicated memory bank storing memory units from previously generated content. By retrieving these memory units from the memory bank and incorporating the information by memory blocks to guide generation, our approach ensures long-term consistency in world simulation.

## 3 WORLDMEM

This section details the methodology of WORLDMEM. Sec. 3.1 introduces the relevant preliminaries, while Sec. 3.2 describes the interactive world simulator serving as our baseline. Sec. 3.3 and 3.4 present the core of our proposed memory mechanism.

### 3.1 Preliminary

**Video diffusion models.** Video diffusion models generate video sequences by iteratively denoising Gaussian noise through a learned reverse process:

$$p_\theta(\mathbf{x}_t^{k-1}|\mathbf{x}_t^k) = \mathcal{N}(\mathbf{x}_t^{k-1}; \mu_\theta(\mathbf{x}_t^k, k), \sigma_k^2 \mathbf{I}), \tag{1}$$

where all frames $(\mathbf{x}_t^k)_{1 \le t \le T}$ share the same noise level $k$, and $T$ is the context window length. This *full-sequence* approach enables global guidance but lacks flexibility in sequence length and autoregressive generation.

**Autoregressive video generation.** Autoregressive video generation aims to extend videos over the long term by predicting frames sequentially (Kondratyuk et al., 2024; Wu et al., 2023). While various methods exist for autoregressive generation, Diffusion Forcing (DF) (Chen et al., 2025) provides a neat and effective approach to achieve this. Specifically, DF introduces *per-frame noise levels* $k_t$:

$$p_\theta(\mathbf{x}_t^{k_t-1}|\mathbf{x}_t^{k_t}) = \mathcal{N}(\mathbf{x}_t^{k_t-1}; \mu_\theta(\mathbf{x}_t^{k_t}, k_t), \sigma_{k_t}^2 \mathbf{I}), \tag{2}$$

Unlike full-sequence diffusion, DF generates video flexibly and stably beyond the training horizon. Autoregressive generation is a special case when only the last one or a few frames are noisy. With autoregressive video generation, long-term interactive world simulation becomes feasible.

### 3.2 Interactive World Simulation

Before introducing the memory mechanism, we first present our interactive world simulator, which models long video sequences using an auto-regressive conditional diffusion transformer. Interaction is achieved by embedding external control signals, primarily actions, into the model through dedicated conditioning modules (Parker-Holder et al., 2024; Decart et al., 2024; Yu et al., 2025c).

Following prior work (Decart et al., 2024), we adopt a conditional Diffusion Transformer (DiT) (Peebles and Xie, 2023) architecture for video generation, and Diffusion Forecasting (DF) (Chen et al.,

2025) for autoregressive prediction. As shown in Figure 2(a), our model consists of multiple DiT blocks with spatial and temporal modules for spatiotemporal reasoning. The temporal module applies causal attention to ensure that each frame only attends to preceding frames.

The actions are injected by first projected into the embedding space using a multi-layer perceptron (MLP). The resulting action embeddings are added to the denoising timestep embeddings and injected into the temporal blocks using Adaptive Layer Normalization (AdaLN) (Xu et al., 2019), following the paradigm of Bar et al. (2024); Decart et al. (2024). In our Minecraft experiments, the action space contains 25 dimensions, including movements, view adjustments, and event triggers. We also apply timestep embeddings to the spatial blocks in the same manner, although this is omitted from the figure for clarity. Standard architectural components such as residual connections, multi-head attention, and feedforward networks are also not shown.

The combination of conditional DiT and DF provides a strong baseline for long-term interactive video generation. However, due to the computational cost of video synthesis, the temporal context window remains limited. As a result, content outside this window is forgotten, which leads to inconsistencies during long-term generation (Decart et al., 2024).

## 3.3 Memory Representation and Retrieval

To address the limited context window of video generative models, we introduce a *memory mechanism* that enables the model to retain and retrieve information beyond the current generation window. This mechanism maintains a *memory bank* composed of historical frames and their associated state information: $\{(\mathbf{x}_i^m, \mathbf{p}_i, t_i)\}_{i=1}^N$, where $\mathbf{x}_i^m$ denotes a memory frame, $\mathbf{p}_i \in \mathbb{R}^5$ (x, y, z, pitch, yaw) is its pose, and $t_i$ is the timestamp. Each tuple is referred to as a *memory unit*. We save $\mathbf{m}_i$ in token-level, which is compressed by the visual encoder but retains enough details for reconstruction. The corresponding states $\{(\mathbf{p}, t)\}$ play a critical role not only in memory retrieval but also in enabling state-aware memory conditioning.

---

**Algorithm 1:** Memory Retrieval Algorithm

---

**Input:** Memory bank of $N$ historical states
$\qquad \{(\mathbf{x}_i^m, \mathbf{p}_i, t_i)\}_{i=1}^N$;
Current state $(\mathbf{x}_c, \mathbf{p}_c, t_c)$; memory condition length $L_M$;
Similarity threshold $tr$; weights $w_o, w_t$.
**Output:** A list of selected state indices $S$
**Compute Confidence Score:**
Compute FOV overlap ratio $\mathbf{o}$ via Monte Carlo sampling.
Compute time difference $\mathbf{d} = \mathrm{Concat}(\{|t_i - t_c|\}_{i=1}^n)$.
Compute confidence $\boldsymbol{\alpha} = \mathbf{o} \cdot w_o - \mathbf{d} \cdot w_t$.

**Selection with Similarity Filtering:**
Initialize $S = \varnothing$
**for** $m = 1$ **to** $L_M$ **do**
$\quad$ Select $i^*$ with highest $\alpha_{i^*}$
$\quad$ Append $i^*$ to $S$
$\quad$ Remove all $j$ where similarity$(i^*, j) > tr$
**return** $S$

---

**Memory Retrieval.** Since the number of memory frames available for conditioning is limited, an efficient strategy is required to sample memory units from the memory bank. We adopt a greedy matching algorithm based on frame-pair similarity, where similarity is defined using the field-of-view (FOV) overlap ratio and timestamp differences as confidence measures. Algorithm 1 presents our approach to memory retrieval. Although simple, this strategy proves effective in retrieving relevant information for conditioning. Moreover, the model's reasoning over memory helps maintain performance even when the retrieved content is imperfect.

## 3.4 State-aware Memory Condition

After retrieving necessary memory units, unlike prior methods that use memory mainly for temporal smoothness (Zheng et al., 2024a) or semantic guidance (Wu et al., 2025b; Rahman et al., 2023), our goal is to explicitly reconstruct previously seen visual content – even under significant viewpoint or scene changes. This requires the model to perform spatiotemporal reasoning to extract relevant information from memory, which we model using cross-attention (Vaswani et al., 2017). Since relying solely on visual tokens can be ambiguous, we incorporate the corresponding states as cues to enable state-aware attention.

**State Embedding.** State embedding provides essential spatial and temporal context for memory retrieval. To encode spatial information, we adopt Plücker embedding (Sitzmann et al., 2021) to convert 5D poses $\mathbf{p} \in \mathbb{R}^5$ into dense positional features $\mathrm{PE}(\mathbf{p}) \in \mathbb{R}^{h \times w \times 6}$, following (He et al., 2024; Gao et al., 2024). Temporal context is captured via a lightweight MLP over sinusoidal embedded

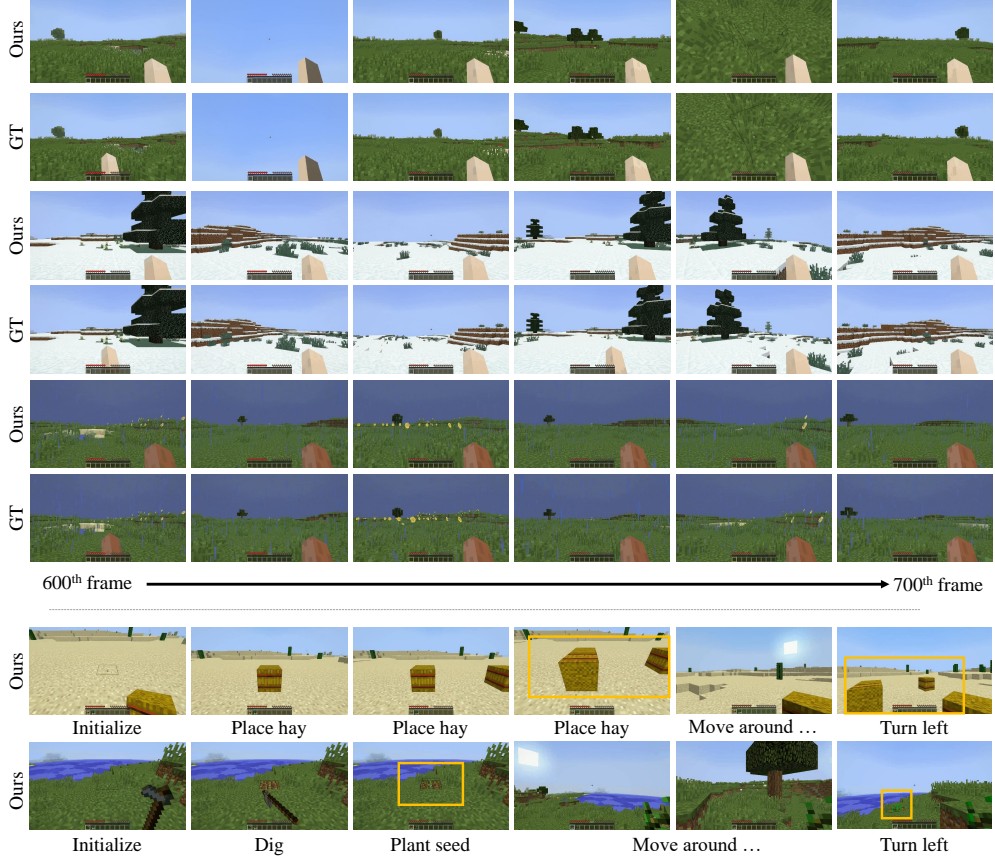

Figure 3: **Qualitative results.** We showcase WORLDMEM's capabilities through two sets of examples. **Top**: A comparison with Ground Truth (GT). WORLDMEM accurately models diverse dynamics (e.g., rain) by conditioning on 600 past frames, ensuring temporal consistency. **Bottom**: Interaction with the world. Objects like hay in the desert or wheat in the plains persist over time, with wheat visibly growing. For the best experience, see the supplementary videos.

(*SE*) timestamps. The final embedding is (Figure 2 (c)):

$$\mathbf{E} = G_p(\text{PE}(\mathbf{p})) + G_t(\text{SE}(t)), \tag{3}$$

where $G_p$ and $G_t$ are MLPs mapping pose and time into a shared space.

**State-aware Memory Attention.** To support reconstruction under viewpoint and temporal shifts, we introduce a state-aware attention mechanism that incorporates spatial-temporal cues into memory retrieval. By conditioning attention on both visual features and state information, the model achieves more accurate reasoning between input and memory.

Let $\mathbf{X}_q \in \mathbb{R}^{l_q \times d}$ denote the flattened feature map of input frames (queries), and $\mathbf{X}_k \in \mathbb{R}^{l_k \times d}$ the concatenated memory features (keys and values). We first enrich both with their corresponding state embeddings $\mathbf{E}_q$ and $\mathbf{E}_k$:

$$\tilde{\mathbf{X}}_q = \mathbf{X}_q + \mathbf{E}_q, \quad \tilde{\mathbf{X}}_k = \mathbf{X}_k + \mathbf{E}_k. \tag{4}$$

Cross-attention is then applied to retrieve relevant memory content and output updated $\mathbf{X}'$:

$$\mathbf{X}' = \text{CrossAttn}(Q = p_q(\tilde{\mathbf{X}}_q), \ K = p_k(\tilde{\mathbf{X}}_k), V = p_v(\mathbf{X}_k)), \tag{5}$$

where $p_q$, $p_k$, and $p_v$ are learnable projections.

To simplify the reasoning space, we adopt a *relative state* formulation. For each query frame, the state is set to a zero reference (*e.g.*, the pose is reset to the identity and the timestamp to zero), while the states of key frames are normalized to relative values. This design, illustrated in Figure 2(d), improves alignment under viewpoint changes and simplifies the learning objective.

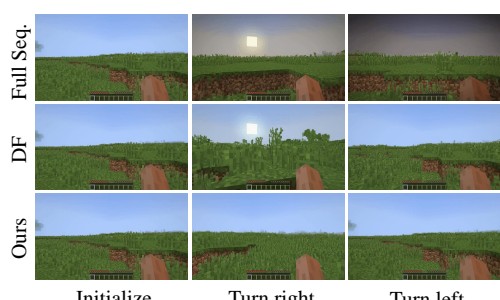
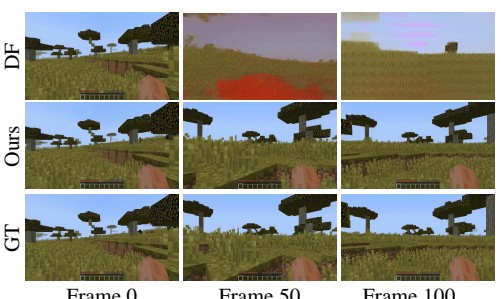

**Figure 4: Within context window evaluation.** The motion sequence involves turning right and returning to the original position, showing self-contained consistency.

**Figure 5: Beyond context window evaluation.** Diffusion-Forcing suffers inconsistency over time, while ours maintains quality and recovers past scenes.

Table 1: Evaluation on Minecraft

| Within context window | | | |
| --- | --- | --- | --- |
| Methods | PSNR ↑ | LPIPS ↓ | rFID ↓ |
| Full Seq. | 20.14 | 0.0691 | 13.87 |
| DF | 24.11 | 0.0094 | 13.88 |
| Ours | **25.98** | **0.0072** | **13.73** |
| Beyond context window | | | |
| Methods | PSNR ↑ | LPIPS ↓ | rFID ↓ |
| Full Seq. | / | / | / |
| DF | 17.32 | 0.4376 | 51.28 |
| Ours | **23.98** | **0.1429** | **15.37** |

Table 2: Ablation on embedding designs

| Pose type | Embed. type | PSNR ↑ | LPIPS ↓ | rFID ↓ |
| --- | --- | --- | --- | --- |
| Sparse | Absolute | 20.67 | 0.2887 | 39.23 |
| Dense | Absolute | 23.63 | 0.1830 | 29.34 |
| Dense | Relative | **23.98** | **0.1429** | **15.37** |

Table 3: Ablation on memory retrieve strategy

| Strategy | PSNR ↑ | LPIPS ↓ | rFID ↓ |
| --- | --- | --- | --- |
| Random | 18.32 | 0.3224 | 47.35 |
| + Confidence Filter | 23.12 | 0.1863 | 24.33 |
| + Similarity Filter | **23.98** | **0.1429** | **15.37** |

**Incorporating memory into pipeline.** We incorporate memory frames into the pipeline by treating them as *clean* inputs during both training and inference. As shown in Figure 2 (a-b), during training, memory frames are assigned the lowest noise level $k_{\min}$, while context window frames receive independently sampled noise levels from the range $[k_{\min}, k_{\max}]$. During inference, both memory and context frames are assigned $k_{\min}$, while the current generating frames are assigned $k_{\max}$.

To restrict memory influence only to memory blocks, we apply a temporal attention mask:

$$A_{\mathrm{mask}}(i, j) = \begin{cases} 1, & i \leq L_M \text{ and } j = i \\ 1, & i > L_M \text{ and } j \leq i \\ 0, & \text{otherwise} \end{cases} \qquad (6)$$

where $L_M$ is the number of memory frames that are appended before frames within the context window. This guarantees causal attention while preventing memory units from affecting each other.

## 4 Experiments

**Datasets.** We use MineDojo (Fan et al., 2022) to create diverse training and evaluation datasets in Minecraft, configuring diverse environments (*e.g.*, plains, savannas, ice plains, and deserts), agent actions, and interactions. For real-world scenes, we utilize RealEstate10K (Zhou et al., 2018) with camera pose annotations to evaluate long-term world consistency.

**Metrics.** For quantitative evaluation, we employ reconstruction metrics, where the method of obtaining ground truth (GT) varies by specific settings. We then assess the consistency and quality of the generated videos using PSNR, LPIPS (Zhang et al., 2018), and reconstruction FID (rFID) (Heusel et al., 2017), which collectively measure pixel-level fidelity, perceptual similarity, and overall realism.

**Experimental details.** For our experiments on Minecraft (Fan et al., 2022), we utilize the Oasis (Decart et al., 2024) as the base model. Our model is trained using the Adam optimizer with a fixed

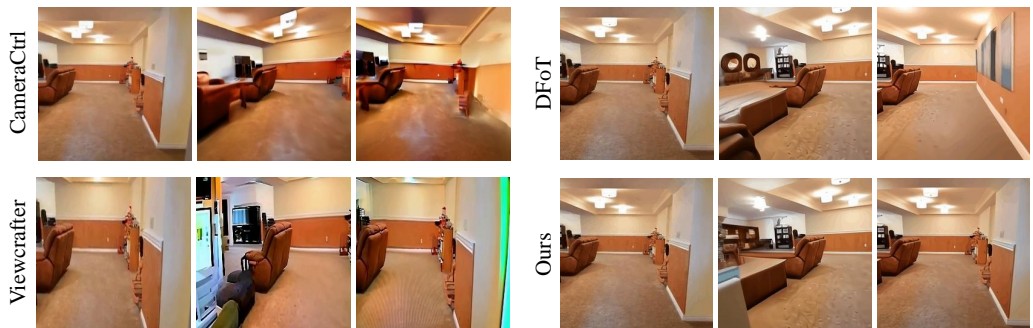

Figure 6: **Results on RealEstate (Zhou et al., 2018).** We visualize loop closure consistency over a full camera rotation. The visual similarity between the first and last frames serves as a qualitative indicator of 3D spatial consistency.

Table 4: Evaluation on RealEstate10K

| Methods | PSNR ↑ | LPIPS ↓ | rFID ↓ |
|---|---|---|---|
| CameraCtrl (He et al., 2024) | 13.19 | 0.3328 | 133.81 |
| TrajAttn (Xiao et al., 2024) | 14.22 | 0.3698 | 128.36 |
| Viewcrafter (Yu et al., 2024c) | 21.72 | 0.1729 | 58.43 |
| DFoT (Song et al., 2025) | 16.42 | 0.2933 | 110.34 |
| Ours | **23.34** | **0.1672** | **43.14** |

learning rate of $2 \times 10^{-5}$. Training is conducted at a resolution of $640 \times 360$, where frames are first encoded into a latent space via a VAE at a resolution of $32 \times 18$, then further patchified to $16 \times 9$. Our training dataset comprises approximately 12K long videos, each containing 1500 frames, generated from Fan et al. (2022). During training, we employ an 8-frame temporal context window alongside an 8-frame memory window. The model is trained for approximately 500K steps using 4 GPUs, with a batch size of 4 per GPU. For the hyperparameters specified in Algorithm 1 of the main paper, we set the similarity threshold $tr$ to 0.9, $w_o$ to 1, and $w_t$ to $0.2/t_c$. For the noise levels in Eq. (5) and Eq. (6), we set $k_{\min}$ to 15 and $k_{\max}$ to 1000.

For our experiments on RealEstate10K (Zhou et al., 2018), we adopt DFoT (Song et al., 2025) as the base model. The RealEstate10K dataset provides a training set of approximately 65K short video clips. Training is conducted at a resolution of $256 \times 256$, with frames patchified to $128 \times 128$. The model is trained for approximately 50K steps using 4 GPUs, with a batch size of 8 per GPU.

### 4.1 Results on Generation Benchmark

**Comparisons on Minecraft Benchmark.** We compare our approach with a standard full-sequence (Full Seq.) training method (He et al., 2024; Wang et al., 2024) and Diffusion Forcing (DF) (Chen et al., 2025). The key differences are as follows: the full-sequence conditional diffusion transformer (Peebles and Xie, 2023) maintains the same noise level during training and inference, DF introduces different noise levels for training and inference, and our method incorporates a memory mechanism. To assess both short-term and long-term world consistency, we conduct evaluations within and beyond the context window. We evaluate both settings on 300 test videos. In the following experiments, the agent's poses are generated by the game simulator as ground truth. However, in real-world scenarios, only the action input is available, and the pose is not directly observable. In such cases, the next-frame pose can be predicted based on the previous scenes, past states, and the upcoming action. We explore this design choice in the supplementary material.

*Within context window.* For this experiment, all methods use a context window of 16, while our approach additionally maintains a memory window of 8. We test on customized motion scenarios (*e.g.*, turn left, then turn right or move forward, then backward) to assess self-contained consistency, where the ground truth consists of previously generated frames at the same positions. As shown in Table 1 and Figure 4, the full-sequence baseline suffers from inconsistencies even within its own context window. DF improves consistency by enabling greater information exchange among generated frames. Our memory-based approach achieves the best performance, demonstrating the effectiveness of integrating a dedicated memory mechanism.

Table 5: Ablation on sampling strategy for training

| Sampling strategy | PSNR ↑ | LPIPS ↓ | rFID ↓ |
|---|---|---|---|
| Small-range | 19.23 | 0.3786 | 46.55 |
| Large-range | 21.11 | 0.3855 | 42.96 |
| Progressive | **23.98** | **0.1429** | **15.37** |

*Beyond context window.* In this setting, all methods use a context window of 8 and generate 100 future frames; our method further employs a memory window of 8 while initializing a 600-frame memory bank. We compute the reconstruction error using the subsequent 100 ground truth frames after 600 frames. Full-sequence methods can not roll out that long so we exclude it. DF exhibits poor PSNR and LPIPS scores, indicating severe inconsistency with the ground truth beyond the context window. Additionally, its low rFID suggests notable quality degradation. In contrast, our memory-augmented approach consistently outperforms others across all metrics, demonstrating superior long-term consistency and quality preservation. Figure 5 further substantiates these findings.

Figure 3 showcases WORLDMEM's capabilities. The top section demonstrates its ability to operate in a free action space across diverse environments. Given a 600-frame memory bank, our model generates 100 future frames while preserving the ground truth's actions and poses, ensuring strong world consistency. The bottom section highlights dynamic environment interaction. By using timestamps as embeddings, the model remembers environmental changes and captures natural event evolution, such as plant growth over time.

**Comparisons on Real Scenarios.** We compare our method with prior works (He et al., 2024; Xiao et al., 2024; Yu et al., 2024c; Song et al., 2025) on the RealEstate10K dataset (Zhou et al., 2018). We design 5 evaluation trajectories, each starting and ending at the same pose, across 100 scenes. The trajectory lengths range from 37 to 60 frames – exceeding the training lengths of all baselines (maximum 25 frames).

CameraCtrl (He et al., 2024), TrajAttn (Xiao et al., 2024), and DFoT (Song et al., 2025) discard past frames and suffer from inconsistency. Viewcrafter (Yu et al., 2024c) incorporates explicit 3D reconstruction, yielding better results, but is constrained by errors in post-processing such as reconstruction and rendering. As shown in Table 4 and Figure 6, our approach achieves superior performance across all metrics. However, the RealEstate dataset inherently limits the full potential of our method, as it consists of short, non-interactive clips with limited temporal complexity. We leave evaluation under more challenging and interactive real-world scenarios for future work.

## 4.2 Ablation

**Embedding designs.** The design of embeddings within the memory block is crucial for cross-frame relationship modeling. We evaluate three strategies (Table 2): (1) sparse pose embedding with absolute encoding, (2) dense pose embedding with absolute encoding, and (3) dense pose embedding with relative encoding. Results show that dense pose embeddings (Plücker embedding) significantly enhance all metrics, emphasizing the benefits of richer pose representations. Switching from absolute to relative encoding further improves performance, particularly in LPIPS and rFID, by facilitating relationship reasoning and information retrieval. As illustrated in Figure 7, absolute embeddings accumulate errors over time, while relative embeddings maintain stability even beyond 300 frames.

**Sampling strategy for training.** We compare different sampling strategies during training in the Minecraft benchmark. Small-range sampling restricts memory conditioning to frames within 2m in the Minecraft world, while large-range sampling extends this range to 8m. Progressive sampling, on the other hand, begins with small-range samples for initial training steps and then gradually expands to large-range samples.

As shown in Table 5, both small-range and large-range sampling struggle with consistency and quality, whereas progressive sampling significantly improves all metrics. This suggests that gradually increasing difficulty during training helps the model learn to reason and effectively query information from memory blocks.

**Time condition.** We ablate the effectiveness of the timestamp condition (for both embedding and retrieval) in Table 6. We curate 100 video samples featuring placing events and evaluate whether future generations align with event progression. As shown in the table, incorporating the time

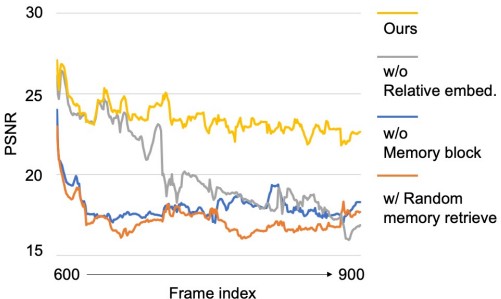

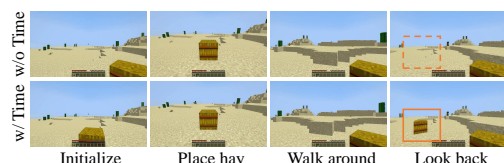

Figure 7: **Long-term Generation Comparison.** This figure presents the PSNR of different ablation methods compared to the ground truth over a 300-frame sequence. The results show that our method without memory blocks or using random memory retrieval exhibits immediate inconsistencies with the ground truth. Additionally, the model lacking relative embeddings begins to degrade significantly beyond 100 frames. In contrast, our full method maintains strong consistency even beyond 300 frames.

Figure 8: **Results w/o and w/ time condition.** Without timestamps, the model fails to differentiate memory units from the same location at different times, causing errors. With time conditioning, it aligns with the updated world state, ensuring consistency.

Table 6: Ablation on time condition

| Time condition | PSNR ↑ | LPIPS ↓ | rFID ↓ |
|:---:|:---:|:---:|:---:|
| w/o | 23.17 | 0.1989 | 23.89 |
| w/ | **25.12** | **0.1613** | **16.53** |

condition significantly improves PSNR and LPIPS, indicating that adding temporal information helps the model faithfully reproduce event changes in world simulation. Since events like plant growth are inherently unpredictable, we do not conduct quantitative evaluations on such cases but instead provide qualitative illustrations in Figure 8.

**Memory retrieve strategy.** We analyze memory retrieval strategies in Table 3. Random sampling from the memory bank leads to poor performance and severe quality degradation, as evidenced by a sharp drop in rFID and rapid divergence from the ground truth (Figure 7). The confidence-based filtering significantly enhances consistency and generation quality. Additionally, we refine retrieval by filtering out redundant memory units based on similarity, further improving all evaluation metrics and demonstrating the effectiveness of our approach.

## 5   Limitations and Future works

Despite the effectiveness of our approach, certain issues warrant further exploration. First, we cannot guarantee that we can always retrieve all necessary information from the memory bank In some corner cases (*e.g.* , when views are blocked by obstacles), relying solely on view overlap may be insufficient. Second, our current interaction with the environment lacks diversity and realism. In future work, we plan to extend our models to real-world scenarios with more realistic and varied interactions. Lastly, our memory design still entails linearly increasing memory usage, which may impose limitations when handling extremely long sequences.

## 6   Conclusion

In conclusion, WORLDMEM tackles the longstanding challenge of maintaining long-term consistency in world simulation by employing a memory bank of past frames and associated states. Its memory attention mechanism enables accurate reconstruction of previously observed scenes, even under large viewpoints or temporal gaps, and effectively models dynamic changes over time. Extensive experiments in both virtual and real settings confirm WORLDMEM's capacity for robust, immersive world simulation. We hope our work will encourage further research on the design and applications of memory-based world simulators.

**Acknowledgements.** This research is supported by the National Research Foundation, Singapore, under its NRF Fellowship Award <NRF-NRFF16-2024-0003>. This research is also supported by NTU SUG-NAP, as well as cash and in-kind funding from NTU S-Lab and industry partner(s).

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
