# OpenReview forum: "WorldMem: Long-term Consistent World Simulation with Memory"
_NeurIPS.cc/2025/Conference — NeurIPS 2025 poster_

### Official Review · Reviewer_sC3T · 2025-06-28

**Clarity:** 3
**Significance:** 4
**Originality:** 3
**Rating:** 5
**Confidence:** 4

**Summary:**

The manuscript introduces a key-frame based memory model for image-based world models. The memory consists of previous state anchored in space via camera pose and time. When generating a new frame, first a set of memory/key frames are retrieved based on viewpoint overlap and relative time. The next frame is generated using these additional frames for context. This approach proves very effective in generating videos beyond the training time horizon and performs better than related works with fixed context windows or that use reconstruction to fuse a memory representation.

**Questions:**

Fig 6: it is hard to tell what the expected frames should be - it would be good to add some GT frames here to anchor and be able to compare.

**Ethical Concerns:**

["NO or VERY MINOR ethics concerns only"]

**Final Justification:**

I have read the rebuttal and the other reviews.

I appreciate the authors clarifications and am happy to stay with my original rating of accept. I did not see anything major in the other reviews either to change my positive rating.

I trust that the authors will include the additional studies on retrieval duration, long generation and ablation of the number of context frames into the publication.

**Limitations:**

limitations are not written in the main paper. The limitations summarized in the supplement are solid. Especially the linear memory complexity. Consider lifting them to the main paper.

**Quality:**

4

**Strengths And Weaknesses:**

Overall I am very excited about this line of work and the very promising results in generating consistent video over long time horizons with the flexible key-frame-based memory setup. I think this is a significant paper due to the simplicity of the approach and the demonstrated performance.

- Strengths:
    - The key-frame-based memory approach is flexible and induces minimal inductive biases. Only the most fundamental quantities of space and time are used. The experiments on 600 frame memory context and generation of 100 frames into the future demonstrate very clearly the value and power of the approach in long term generation.
    - The qualitative evidence that temporal dynamics like growing grass can be generated based on the world memory is exciting. It will be very interesting to see if more intricate dynamics can be generated as well in future work.
    - The paper is well and clearly written with good illustrations to help convey the message and the approach. Illustrating video generations in picture captions is hard. The supplemental videos are really helpful in this regard.
    - The experiments both quantitatively and qualitatively support the claims and design choices well. In particular the long-term generation beyond the context length is a powerful signal (Tab 1 and Fig 7).
- Weaknesses:
    - One of the main limitations and potential weaknesses for very long time horizon generation of this approach is the linear memory growth since all past frames are stored into the memory bank. Reconstruction-based approaches can theoretically overcome this limitation but as shown in the paper lead to lower quality generations.
    - Currently WorldMem gets 16+8 frame input and related work gets only 16. This makes it not 100% comparable. I think we need to see an ablation where WorldMem gets 8+8 frames input to get the same number of frames. This would clearly show the difference of getting 8 frames from the memory bank vs getting just 8 frames from the past (other methods).
    - I would have appreciated a little more detail on the per frame noise levels used for generation and how those affect the outcome.

---

> ### Author Rebuttal · Authors · 2025-07-30
>
> We thank the reviewer for the thoughtful feedback and helpful suggestions regarding fairness of comparisons, memory scaling, and evaluation clarity. We will revise the paper accordingly.
>
> ---
>
> ### **W1: Linear memory growth**
>
> We agree that the memory bank grows linearly with video length. However, in practice, we find the cost to be manageable:
>
> | Number of Memory Candidates | Retrieval Time (s) |
> |------------------------|--------------------|
> | 10                     | 0.04               |
> | 100                    | 0.06               |
> | 600                    | 0.10               |
> | 1000                   | 0.16               |
>
> The memory bank itself is lightweight. For example, storing 600 visual memory tokens of shape `[600, 16, 18, 32]` in `float32` takes only about **21MB**. Furthermore, since memory tokens are detached and not used in backpropagation during inference, the actual GPU memory footprint is low. Retrieval time increases slowly with memory size, making our method scalable in practice even for long sequences.
>
> ---
>
> ### **W2: Input length mismatch in comparison**
>
> Thank you for pointing this out. While our method uses 16 context frames plus 8 retrieved memory frames, other baselines typically use 16 continuous past frames. We agree this makes direct comparison not entirely fair. However, it is worth noting that the primary role of context frames in our method is to maintain short-term temporal smoothness, rather than to provide rich content information, since we have memory as context.
>
> We provide the following ablation to show how the number of context frames affects performance. The results suggest that as long as the context includes a few relevant frames (e.g., more than 3), the generation quality remains stable:
>
> | Number of Context Frames | PSNR  |
> |---------------------|--------|
> | 1                   | 23.12  |
> | 4                   | 27.09  |
> | 8                   | 26.84  |
> | 16                  | 27.01  |
>
>
>
> ---
>
> ### **W3: Per-frame noise level**
>
> We adopt the standard diffusion forcing setup, where each frame is trained with independently sampled noise levels. During training, we do not perform specific ablations to optimize per-frame-level noise strategies. Instead, we follow the default setting where each frame is independently perturbed with randomly sampled noise levels. This setup simplifies training and encourages the model to generalize across a range of noise conditions.
>
> Our main motivation for using per-frame noise levels is to enable more flexible control during inference. In particular, we apply an **autoregressive inference strategy**, where frames are denoised sequentially.
>
>
> ---
>
> ### **Q1: Missing ground truth in Figure 6**
>
> Thank you for the suggestion. As this is a synthetic rotation trajectory not observed in training data, there is no ground truth to compare with. Instead, the goal of Fig. 6 is to visualize loop closure consistency over one full rotation, where the visual similarity between the first and last frames provides a qualitative indicator of 3D spatial consistency. We will make this clearer in the revised version.
>
> ---
>
> ### **Limitations missing in main paper**
>
> We appreciate this suggestion. We will lift this section into the main paper in the revised version.
>
>
>
> ---
>
> We hope our responses have addressed your concerns. If there are any unclear points or remaining questions, we would be happy to further clarify and discuss.

---

> > ### Comment · Reviewer_sC3T · 2025-08-06
> >
> > I have read the rebuttal and the other reviews.
> >
> > I appreciate the authors clarifications and am happy to stay with my original rating of accept. I did not see anything major in the other reviews either to change my positive rating.
> >
> > I trust that the authors will include the additional studies on retrieval duration, long generation and ablation of the number of context frames into the publication.

---

### Official Review · Reviewer_7GuY · 2025-06-29

**Clarity:** 2
**Significance:** 3
**Originality:** 3
**Rating:** 4
**Confidence:** 4

**Summary:**

this paper proposes a method to use past visual and state information as a kind a memory to generated videos with improved content consistency. useful information is retrieved and injected through cross-attention.

**Questions:**

1. this paper proposed to use cross-attention to make use of the memory information, which treats the retrieved memory as condition (like input text). what if the retrieved memory is incomplete or flawed? i suspect using cross-attention may cause the whole generation model to be sensitive to errors of memory retrieval. please add more analyses on the pros and cons of using the cross-attention mechnism.
2. figure 2(d) is not clear. in the "State-aware Memory Attention" part, how the cross attention is performed when clear frames within the context window and clear frames within the memory bank is the input?

**Ethical Concerns:**

["NO or VERY MINOR ethics concerns only"]

**Final Justification:**

the rebuttals addressed my concerns to a large extent. i will keep my original rating.

**Limitations:**

yes

**Quality:**

3

**Strengths And Weaknesses:**

strength:
this paper is well-written and easy to follow, and the proposed method does alleviates the content consistency problem in the experiment settings of this paper.

weaknesses:
in the experiments, the generated video has quite limited length - 16 to 100 frames - which is not a typical scenario for showing the content consistency issue for long video generation (minecraft game video in this paper). for longer (minute level) video generation, how will the performance of the propose method be?

---

> ### Author Rebuttal · Authors · 2025-07-30
>
> We sincerely thank the reviewer for the valuable suggestions and sharp observations. Your comments on long-horizon generation, robustness of memory conditioning, and architectural clarity are greatly appreciated. We will revise the paper to include more details.
>
> ---
>
> ### **W1: Limited generation length**
>
> To further assess long-range consistency, we provide the following PSNR results for longer sequences:
>
> | Number of Generated Frames | PSNR  |
> |-------------|--------|
> | 100         | 23.1   |
> | 300         | 22.8   |
> | 500         | 22.7   |
> | 800         | 21.4   |
> | 1000        | 20.8   |
>
> These results indicate that our method maintains strong reconstruction quality up to 500 frames, and although quality degrades at 1000 frames, the generation does not collapse and remains structurally reasonable. This demonstrates that our method is capable of handling relatively long sequences, beyond the typical short-range settings explored in prior work.
>
>
>
> ---
>
> ### **Q1: On the robustness and design of memory cross-attention**
>
> Thank you for the thoughtful question. Our use of cross-attention to incorporate memory is motivated by a balance between robustness and efficiency.
>
> - **Robustness to noisy memory**: During training, memory is sampled randomly from large video pools, which often include unrelated frames. The model thus learns to selectively attend to informative content while ignoring irrelevant memory. This naturally increases robustness to retrieval imperfections.
>
> Using cross-attention for memory conditioning is not strictly necessary. One could alternatively apply full self-attention over both the current frame and the memory. However, we adopt cross-attention for the following reasons:
>
> - It introduces a clear directional inductive bias, where the current denoising frame attends to the memory without allowing information to flow back.
> - It significantly reduces computational and memory cost, particularly when the context or memory length is large.
>
> Thus, the training and inference would both be more effective and efficient.
>
> That said, cross-attention is less standard than full self-attention in typical Transformer architectures. Full self-attention offers greater capacity and flexibility for learning rich interactions between memory and context, which might be suitable for large-scale training.
>
>
>
>
> ---
>
> ### **Q2: Clarification on Figure 2(d)**
>
> We appreciate your comments. In the "State-aware Memory Attention" module, we use all tokens, including clean frames from the context window, clean frames from the memory bank, and noisy frames, as queries. The clean frames within the memory bank are used as keys and values in the cross-attention operation. We will revise Figure 2(d) to highlight the query-key-value assignments more clearly, including visual markers for which tokens are used in each role.
>
>
> ---
>
>
> We hope our responses have addressed your concerns. If there are any unclear points or remaining questions, we would be happy to further clarify and discuss.

---

> > ### Comment · Reviewer_7GuY · 2025-08-04
> >
> > thanks for the rebuttal.
> >
> > i still am not clear about the specific design of the "State-aware Memory Attention" module. the authors mentioned that "all tokens are used as queries and the clean frames within the memory bank are used as keys and values", then why is it necessary to make memory tokens self-attend in the cross-attention memory block? and what exactly happens in spatial blocks and temporal blocks?

---

> > > ### Author Response · Authors · 2025-08-05
> > >
> > > Thank you for your questions.
> > >
> > > We allow memory tokens to self-attend because they are treated as the same type of tokens as the regular ones. They are simply cleaner and contain less noise. To maintain consistency across the pipeline, memory tokens are processed with the same operations as other tokens. Skipping self-attention for memory tokens could introduce a distributional mismatch or representation gap between them and the rest of the tokens.
> > >
> > > Spatial and temporal blocks follow standard attention architectures.
> > >
> > > - In the **spatial blocks**, all tokens perform self-attention within their own local frame windows to model intra-frame spatial dependencies.
> > >
> > > - In the **temporal blocks**, we apply inter-frame token-level attention to capture temporal dependencies across frames. Additionally, we introduce a causal attention mask to enforce temporal causality within the context window. We also use a memory mask to prevent memory tokens from attending to other masked tokens. This design ensures that memory frames do not interfere with other frames during this stage, as described in Eq. (6).
> > >
> > > Please tell me if there are further things to be clarified. I will revise this part to make it clear.

---

> > > > ### Comment · Reviewer_7GuY · 2025-08-05
> > > >
> > > > so the memory tokens take sparse self attention in spatial blocks and temporal blocks, and dense self attention in memory blocks? if the authors want to treat "memory tokens are processed with the same operations as other tokens", perhaps a better way is avoid using cross attention? in my opinion, such design of memory usage is just one possible approach but is hardly a sufficiently good approach.

---

> > > > > ### Author Response · Authors · 2025-08-05
> > > > >
> > > > > Thanks for the suggestion. I think you bring up a very reasonable point. As I mentioned in Q1, using cross-attention for memory conditioning is not strictly necessary. In fact, under more scalable settings, adopting unified self-attention across memory and context could be more standard and potentially yield better results.
> > > > >
> > > > > However, under limited resources, our current decomposed design is both **efficient** and **effective** during training and inference. By introducing cross-attention, we avoid densely attending across all frames within the context. This not only reduces the computational load but also simplifies the learning objective by introducing a clear directional inductive bias. It is also consistent with our statement that memory tokens are **processed with the same operations as other tokens** since they all attend to the same tokens in the cross attention.
> > > > >
> > > > > Moreover, our proposed **State-aware Memory Attention** is intended to enhance the reasoning ability within attention, where self-attention and cross-attention are simply implementation choices, not the core of the design. The key idea is to encode and utilize the "state" to inform memory access.
> > > > >
> > > > > Hope this helps clarify our intention, happy to continue the discussion!

---

> > > > > > ### Comment · Reviewer_7GuY · 2025-08-06
> > > > > >
> > > > > > thanks for the response from the authors.
> > > > > >
> > > > > > i still have concerns on the near-linear performance degradation when generating longer videos (as shown in the rebuttal), which cannot be seen as good or reasonable. please elaborate.
> > > > > >
> > > > > > i also have concerns on using additional memory window when comparing with other methods. please elaborate.

---

> > > > > > > ### Author Response · Authors · 2025-08-06
> > > > > > >
> > > > > > > Thank you for your thoughtful questions.
> > > > > > >
> > > > > > > Regarding the first concern, we acknowledge that our method does not completely solve the problem of error accumulation, which remains a fundamental challenge in autoregressive long video generation. However, we would like to highlight that our approach is able to maintain consistent visual quality and semantics over a relatively long duration (e.g., up to one minute), which represents a substantial improvement compared to prior works. We agree this is not a complete solution, and we see this as a promising direction for further research.
> > > > > > >
> > > > > > > As for the use of an additional memory window during evaluation, our design aims to test the model’s ability to leverage long-term context effectively. While this introduces extra context, we ensure that the memory is constructed only from past frames (within a fixed temporal budget) and does not include any future information. We are also careful to make fair comparisons: when reporting performance against other methods, we use consistent evaluation setups or clearly mark the differences in the experiments.
> > > > > > >
> > > > > > > Please let us know if further clarification is needed.

---

> > > > > > > > ### Comment · Reviewer_7GuY · 2025-08-06
> > > > > > > >
> > > > > > > > the rebuttals addressed my concerns to a large extent. i will keep my original rating.

---

### Official Review · Reviewer_xLdM · 2025-06-29

**Clarity:** 3
**Significance:** 3
**Originality:** 2
**Rating:** 4
**Confidence:** 4

**Summary:**

This paper proposes to integrate a memory bank with a video model to enable more consistent
long-horizon video simulation. To do this, the  authors construct a memory bank of past
poses, times and images that the agent has seen. At generation time, relevant frames to the current
location of the agent is achieved through FoV overlap, which are then used to render future frames.

**Questions:**

1) How would the approach illustrated in the paper work for dynamically changing scenes? I.e. in minecraft, there is a zombie that is moving around and disappears when the sun comes out. While the timestep conditioning could help model dynamics for a short amount of time, it seems like a more explicit dynamics model might be needed for large scale dynamic change, where an entity will move from its viewing context in the memory to a very different frame.
2)  Since the paper is based off diffusion forcing, would the history guidance method in [1] also help improve the performance of the method?
3)  The paper uses FoV overlap to compute relevant frames in a memory bank to retrieve. However, in many settings, we may not know the current pose of the agent and the past frames may not form a consistent world model (imagine learning a world model for the wrist camera of a moving robot) How would the authors imagine generalizing their method in this case?

[1] Song et al. History-Guided Video Diffusion.

**Ethical Concerns:**

["NO or VERY MINOR ethics concerns only"]

**Final Justification:**

I remain positive about the paper.

**Limitations:**

I did not find any discussion of the limitations of the approach. One big limitation might be that the retrieval mechanism is likely significantly slower than directly generating video.

**Paper Formatting Concerns:**

Paper formatting looks good.

**Quality:**

2

**Strengths And Weaknesses:**

Strengths
- The paper studies an important and timely problem
- The results seem significant
- The method is clearly explained and well motivated

Weaknesses
- The memory bank selector is hard-coded, preventing the approach from effectively working in settings where we do know the ground truth pose of the agent and where there isn't a well defined 2D map of the scene (i.e. in a multilevel house, the FoV overlap wouldn't make sense because multiple floors would have the same overlap)
- All the results in the paper are done either in Minecraft or on small baseline Realestate 10k images. It would be good to illustrate the applicability of the method on more real world scenes (it seems the method is heavily limited by the need of knowing precise camera poses in a 2D floormap situation)
- All results only show the visual generation results, without any illustrations of the application of the method to downstream decision-making tasks. For instance, can you use the model with MPC to make decisions in minecraft?
- There is a lot of work in interactive world simulation and the authors have cited a limited number of works in the area. The authors should try to cite a more comprehensive set of existing works, for instance, [1] should be discussed.
- The results on the website aren't fully 3D consistent (i.e. in Minecraft, the terrain of the ground is changing)

[1] Yang et al. Learning Interactive Real World Simulators

---

> ### Author Rebuttal · Authors · 2025-07-30
>
> We thank the reviewer for the thoughtful and constructive feedback. We appreciate the detailed analysis and helpful suggestions on memory design, downstream applications, and generalization. We will revise the paper accordingly.
>
> ---
>
> ### **W1: The memory bank selector is hard-coded**
>
> We agree that our current FoV-overlap-based memory retrieval is heuristic and may not generalize well to complex 3D environments (e.g., multi-level buildings). This limitation is acknowledged in the paper (Line 2-5 in Supp.). Nevertheless, it serves as a proof-of-concept showing that structured spatial priors can support long-term consistent world modeling. Despite its simplicity, the approach works well in controlled settings such as Minecraft and indoor scans.
>
> ---
>
> ### **W2: Limited applicability to real-world scenes**
>
> Our current experiments focus on synthetic and small-scale real scenes, mainly due to resource constraints. However, our framework is conceptually applicable to real-world scenarios as long as sufficient sensory input and temporal alignment are available. While relying on known camera poses may seem restrictive, such information is available in many practical settings, such as gaming, robotics, and embodied scanning, via built-in APIs or SLAM systems. As pose estimation tools (e.g., Vggt [1]) continue to improve, this constraint becomes increasingly manageable. We also plan to explore retrieval based on learned visual representations to further reduce dependence on explicit pose information.
>
> ---
>
> ### **W3: No illustration of downstream decision-making tasks**
>
> Thank you for the suggestion. Although our current work focuses on generating quality and consistency, our model is naturally compatible with downstream decision-making frameworks such as Model Predictive Control (MPC). By simulating temporally grounded future observations, our model can be used to evaluate outcomes of candidate action sequences and support action selection. For example, in Minecraft, this allows integration into planning and control pipelines. While we have not demonstrated this in the current work, we consider it an important direction for future research.
>
> ---
>
> ### **W4: Missing related work in interactive world simulation**
>
> We appreciate the reviewer highlighting this. We will extend our related work section to include recent contributions on interactive and long-context video world models, including but not restricted to [2,3,4,5].
>
> ---
>
> ### **W5: Incomplete 3D consistency in generation**
>
> We acknowledge that some generation results, especially in long sequences, may not achieve perfect 3D consistency (e.g., terrain deformation). This remains a general challenge in world modeling. Nonetheless, to the best of our knowledge, our method achieves state-of-the-art consistency compared to existing methods, including very recent approaches [6,7]. We are actively exploring ways to further improve consistency.
>
> ---
>
> ### **Q1: Modeling dynamic scene changes**
>
> This is an important and challenging direction. Our current experiments do not include moving entities (e.g., zombies) primarily due to the computational cost to train a video model that can generate such dynamics with high quality. However, our model architecture is compatible with such dynamics. Since each memory entry is associated with a timestamp, the model can learn temporal evolution, e.g., understanding that a walking zombie at time $t_i$ should appear in a different location at time $t_{i+n}$. This is a key motivation behind our design of *state-aware memory*, which we plan to explore further with dynamic scenes in future work.
>
> ---
>
> ### **Q2: Integration of history guidance**
>
> Yes, history guidance [8] techniques are complementary to our method and can improve temporal coherence and visual fidelity. In our RealEstate10K experiments, we have already applied history guidance to stabilize generation (Table 4 in the main paper). These techniques are orthogonal and can be integrated seamlessly to further enhance results.
>
> ---
>
> ### **Q3: Generalization beyond pose-based retrieval**
>
> Our method leverages the broader concept of *state-aware memory*, with pose and timestep serving as a specific instantiation of the state. In scenarios where explicit poses are unavailable (e.g., egocentric wrist cameras), the same principle applies: define the state (e.g., the position and the rotation angle of robotic arms, past visual input), and learn associations between current states and stored memory. With sufficient training data, we believe a learned retrieval function can effectively generalize to such unstructured environments.
>
> ---
>
> ### **Limitation: Retrieval latency vs. direct generation**
>
> Thank you for highlighting this limitation. While retrieval does introduce some latency, it remains minor even at scale. Below is the empirical timing on an H100 GPU for retrieving 8 memory frames from different memory pool sizes:
>
> | Number of Memory Candidates | Retrieval Time (s) |
> |------------------------|--------------------|
> | 10                     | 0.04               |
> | 100                    | 0.06               |
> | 600                    | 0.10               |
> | 1000                   | 0.16               |
>
> The cost of generation (20 denoising steps) is ~0.9 seconds per frame (excluding retrieval). Thus, retrieval contributes only ~10–20% of the inference time, even when accessing 1000 candidates.
>
> We believe this is a worthwhile trade-off, as retrieval enhances long-term coherence by anchoring generation in relevant context, which pure autoregressive or diffusion models struggle to maintain. We will move this limitation and analysis from the supplementary to the main paper as suggested.
>
>
> ---
> We hope our responses have addressed your concerns. If any points remain unclear, we would be happy to further clarify and discuss.
>
> ---
>
> [1] Wang J, Chen M, Karaev N, et al. Vggt: Visual geometry grounded transformer.
>
> [2] Yang M, Du Y, Ghasemipour K, et al. Learning interactive real-world simulators.
>
> [3] Yu J, Qin Y, Che H, et al. A survey of interactive generative video.
>
> [4] Beattie C, Leibo J Z, Teplyashin D, et al. Deepmind lab.
>
> [5] Hafner D, Pasukonis J, Ba J, et al. Mastering diverse domains through world models.
>
> [6] Chen T, Hu X, Ding Z, et al. Learning World Models for Interactive Video Generation.
>
> [7] Po R, Nitzan Y, Zhang R, et al. Long-context state-space video world models.
>
> [8] Song K, Chen B, Simchowitz M, et al. History-guided video diffusion

---

> > ### Comment · Reviewer_xLdM · 2025-08-06
> > **Final Decision**
> >
> > I thank the authors for the rebuttal and maintain my positive score.

---

### Official Review · Reviewer_Zecv · 2025-07-03

**Clarity:** 3
**Significance:** 3
**Originality:** 3
**Rating:** 5
**Confidence:** 3

**Summary:**

This paper presents a novel framework that incorporates a memory bank consisting of state-aware memory cells.  By modeling temporal and spatial states, this approach aims to address the challenge of maintaining long-term consistency in world model video prediction.  The core of the framework utilizes a specially designed memory attention mechanism that enables the memory unit to store not only information from memory frames, but also pose and timestamp information.  The experimental results in virtual scenes are impressive, with PSNR, LPIPS, and rFID metrics evaluated on long sequences across different scenes and interaction modes in the Minecraft game, achieving state-of-the-art (SOTA) performance.  Additionally, the framework demonstrates promising application prospects in the gaming field.  Similarly, on the real-world RealEstate10K dataset, the approach also achieves SOTA performance.

**Questions:**

1. It would be beneficial to include additional results across a wider range of styles and scene types in both virtual and real-world environments, in order to further demonstrate the generality and robustness of the proposed method.

2. The paper would be strengthened by explicitly stating the practical limitations of the method, including but not limited to memory usage, training time, and inference speed, to better illustrate its real-world applicability.

3. A clear analysis of failure cases, along with a discussion of the underlying reasons for the method's limitations in these scenarios, would provide valuable insight and help guide future improvements.

**Ethical Concerns:**

["NO or VERY MINOR ethics concerns only"]

**Final Justification:**

The authors introduce a compelling framework based on a state-aware memory attention mechanism, which enhances long-term world generation by explicitly modeling temporal and spatial states, providing a solution to the consistency problem.

**Limitations:**

yes

**Quality:**

3

**Strengths And Weaknesses:**

Strengths:

1: This paper is well-written and clearly articulates the inherent challenge of long-term 3D spatial inconsistency in world generation, which effectively motivates the proposed approach and establishes a clear research direction. The authors introduce a compelling framework based on a state-aware memory attention mechanism, which enhances long-term world generation by explicitly modeling temporal and spatial states, providing a solution to the consistency problem.

2: The method demonstrates strong performance both quantitatively and qualitatively, with comprehensive metrics including PSNR, LPIPS, and rFID that validate the effectiveness of the approach.

3: The visual results are impressive and provide clear evidence of the method's capabilities, showing substantial improvements in maintaining visual quality over extended sequences.

4: The method shows clear advantages in maintaining consistency over long temporal horizons, which is the core challenge the paper aims to address and represents a significant advancement in the field.

Weaknesses:

1: The paper lacks a comprehensive analysis of memory usage and the maximum duration for which high-quality reconstructions can be maintained.   Without this analysis, it remains unclear what the practical limitations and scalability boundaries of the proposed method are, which is crucial for evaluating its real-world applicability.

2: The encoding methods employed for pose and timestamp information appear to follow conventional approaches without significant innovation.   The authors should provide stronger motivation for these design choices or explore more sophisticated encoding schemes that could better capture the temporal-spatial relationships.

3: The experimental evaluation of dynamic scene reconstruction is somewhat limited in scope.   For virtual environments, the demonstration is confined to selected Minecraft scenarios, while real-world evaluation focuses primarily on indoor scenes.   A more diverse set of scenarios, including outdoor environments and different types of dynamic content, would strengthen the generalizability claims of the proposed framework.

---

> ### Author Rebuttal · Authors · 2025-07-30
>
> We thank the reviewer for the detailed and insightful comments. Your feedback on memory usage analysis, encoding strategies, and scenario coverage has helped us better identify the limitations of our current design. We will revise the manuscript accordingly.
>
> ---
>
> ### **W1 & Q2: Lack of analysis on memory usage, temporal span, and practical constraints**
>
> We appreciate the reviewer’s concern regarding scalability and real-world applicability. Below, we provide quantitative analysis covering memory usage, generation duration, training cost, and inference efficiency.
>
> - **Memory usage of the memory bank**:
>   The memory bank is lightweight. Storing 600 visual memory tokens with shape `[600, 16, 18, 32]` in `float32` only takes approximately **21MB**.
>
> - **Generation duration**:
>   Our method maintains high-quality reconstruction over long sequences. Below are PSNR values at various lengths:
>
>     | Number of Generated Frames | PSNR  |
>   |-------------|-----------|
>   | 100         | 23.1   |
>   | 300         | 22.8   |
>   | 500         | 22.7   |
>   | 800         | 21.4   |
>   | 1000        | 20.8   |
>
>   These results indicate that our method maintains strong reconstruction quality up to 500 frames, and while degradation is observed at 1000 frames, the generation remains structurally coherent without collapse. This demonstrates that our method is capable of handling relatively long sequences, beyond the typical short-range settings explored in prior work.
>
> - **Retrieval latency** (for 8 memory frames), as memory bank scales:
>
>     | Number of Memory Candidates | Retrieval Time (s) |
>     |------------------------|--------------------|
>     | 10                     | 0.04               |
>     | 100                    | 0.06               |
>     | 600                    | 0.10               |
>     | 1000                   | 0.16               |
>
>   The cost of generation (20 denoising steps) is ~0.9 seconds per frame (excluding retrieval). Thus, retrieval contributes only ~10–20% of the inference time, even when accessing 1000 candidates.
>
>
>
> - **Training setting**:
>   - ~500K training steps to converge on 4×H100 GPUs
>   - Batch size: 4 (the original paper incorrectly reports this and will be corrected)
>   - GPU memory usage per GPU: ~51GB
>   - Total training time: ~4 days
>
> - **Cost Compared with Baseline**
>
>   To further address the reviewer’s concern regarding cost and practical feasibility, we include a detailed comparison below between our method and the baseline (without memory), under consistent settings.
>
>   **Setup**:
>   - 8 context frames + 8 memory frames
>   - Inference uses 20 denoising steps
>   - No acceleration techniques (e.g., timestep distillation, sparse attention)
>   - Single H200
>
>   |                 | **Training**                 |                        | **Inference**              |                        |
>   |-----------------|------------------------------|------------------------|----------------------------|------------------------|
>   |                 | Memory Usage                 | Speed (it/s)          | Memory Usage               | Speed (it/s)          |
>   | **w/o memory**  | 33 GB                        | 3.19                   | 9 GB                       | 1.03                   |
>   | **with memory** | 51 GB                        | 1.76                   | 11 GB                      | 0.89                   |
>
>   Introducing memory increases the computational cost during training, with a moderate rise in memory usage and a reduction in training speed. However, the overhead during inference is relatively minor, with only a small increase in memory usage (from 9 GB to 11 GB) and a slight drop in speed (from 1.03 it/s to 0.89 it/s). Overall, the additional cost introduced by memory is acceptable, particularly during inference, where the impact is minimal.
>
>   Moreover, with modern acceleration techniques, such as timestep distillation, early exit, or sparse attention, the inference speed can be further improved to real-time levels (i.e., approximately 10 FPS), making the approach practical for deployment.
>
>
>
> ---
>
> ### **W2: Conventional encoding of pose and timestamp**
>
> We intentionally adopt standard encoding schemes to represent camera pose and timestamp. This choice ensures better generalization across environments and reduces inductive bias, making our framework more adaptable. While these encodings perform well in our current setting, we agree that more expressive alternatives may further improve performance. Also, such complex designs may incur significantly higher training and inference costs. We consider more sophisticated designs a promising direction for future exploration.
>
> ---
>
> ### **W3 & Q1: Limited dynamic and real-world scenario evaluation**
>
> Our current experiments focus on synthetic environments and small-scale real-world scenes, primarily due to resource limitations. Nonetheless, our framework is conceptually applicable to real-world scenarios, provided that sufficient sensory input and temporal alignment are available.
>
> While our current setup does not include dynamic elements such as moving entities (e.g., zombies), this omission is mainly due to the significant computational cost required to train a video model capable of generating such dynamics with high fidelity. Importantly, our model architecture is inherently compatible with dynamic content. Each memory entry is timestamped, enabling the model to learn temporal evolution. For instance, recognizing that a walking zombie at time  $t_i$ should appear at a different location at time $t_{i+n}$​. This capability is a core motivation behind our design of state-aware memory, which we plan to further explore in future work involving dynamic scenes.
>
> We agree that demonstrating results on more diverse environments, including outdoor and highly dynamic scenes, would strengthen our claims. We consider this an important direction for future work and are working toward scaling the framework to support such settings.
>
> ---
>
> ### **Q3: Failure cases and limitations**
>
> We appreciate the reviewer’s suggestion and will include a dedicated limitations and failure case section in the revised paper. In fact, we have discussed several of these issues in the supplementary and will lift them into the main paper. Below, we summarize key failure modes observed in our current system:
>
> - **Retrieval under occlusion**: Our current memory retrieval strategy is based on view overlap, assuming that overlapping FoVs imply semantic relevance. However, this assumption breaks in the presence of occluders. For example, if the current viewpoint faces a wall, the retrieved frames with similar FoVs may be located behind the wall and thus contain irrelevant content. Our method does not yet model occlusion-aware visibility, which limits retrieval accuracy in cluttered environments.
>
> - **Long-horizon generation degradation**: While our method supports relatively long sequences (e.g., up to 1000 frames), it is still subject to common issues in autoregressive generation, such as quality degradation and error accumulation. As the sequence length increases, we observe texture blurring, loss of spatial detail, and occasional color artifacts (e.g., patchy or unnatural regions), especially beyond 1000 frames. These issues are consistent with failure modes observed in other long-range video generation systems.
>
> We consider addressing these limitations in future work, such as incorporating visibility-awareness into memory retrieval and exploring strategies to improve long-horizon consistency.
>
> ---
>
> We hope our responses have addressed your concerns. If any points remain unclear, we would be happy to further clarify and discuss.

---

> > ### Comment · Reviewer_Zecv · 2025-08-06
> >
> > I have carefully read the authors' rebuttal and appreciate the detailed and thoughtful responses. My original concerns primarily related to the scalability, memory usage, encoding design choices, and the diversity of evaluated scenarios. The authors have addressed these points comprehensively.

---

### Comment · Area_Chair_mEFZ · 2025-08-06
**Please acknowledge and respond to the author rebuttal as soon as possible**

Dear reviewers,

For those who have not yet done so: please acknowledge and respond to the author rebuttal as soon as possible, so that they still might have time to respond.

Kind regards,

AC.

---

### Note · Authors · 2025-08-13

We sincerely thank all reviewers for their time, effort, and constructive feedback. We are encouraged by the positive reception and the recognition of our work’s contributions.

We appreciate Reviewer Zecv for highlighting our clear articulation of the long-term 3D spatial consistency challenge, the novelty of our state-aware memory attention framework, and its strong performance in both virtual and real-world benchmarks. We thank Reviewer xLdM for recognizing the importance and timeliness of the problem, the clarity of our motivation, and the significance of our results. We are grateful to Reviewer sC3T for praising the flexibility of our key-frame-based memory design, its minimal inductive bias, and the strong evidence supporting its effectiveness beyond training horizons. We also thank Reviewer 7GuY for acknowledging the clarity of our writing and the effectiveness of our approach in alleviating content consistency issues.

We acknowledge the valuable suggestions regarding memory usage analysis, pose/timestamp encoding, broader dynamic and real-world evaluation, robustness of memory conditioning, fairness in comparisons, and integration with downstream decision-making tasks. We will revise the manuscript to incorporate these improvements, including lifting limitation discussions into the main paper, expanding evaluations, and clarifying architectural details.

Our vision is to push the boundaries of world simulation toward open-world, interactive, and dynamic scenarios. We aim to extend state-aware memory to richer state definitions, scale to more diverse environments, and integrate seamlessly with decision-making frameworks, ultimately enabling agents to perceive, remember, and interact in complex, evolving worlds over extended time horizons.

---

### Decision · Program_Chairs · 2025-09-17

**Decision:**

Accept (poster)

**Comment:**

The authors introduce an approach to view generation for longer-sequence world modelling, in which a memory bank is used of past frames and states (including time and pose). The approach reaches state-of-the-art performance on MineDojo and RealEstate.

Multiple issues were raised and addressed to the satisfaction of the reviewers, including the scalability of the method in terms of memory and retrieval latency, and the generation of longer-term videos. Other issues are rightly delegated to future work, such as the application of the method to more diverse worlds or more dynamic scenes.

Wit the incorporation of a more elaborate discussion of the limitations in the main manuscript (scaling limitations, dependence on ground-truth) and clarifications on the attention modules and noise properties, this should form a valuable contribution to NeurIPS.